Correlation between fruit weight and nutritional metabolism during development in CPPU-treated Actinidia chinensis ‘Hongyang’

Qiu Guo-liang 1
Zhuang Qi-guo 2
Li Yu-fei 1
Li Si-yu 1
http://orcid.org/0000-0001-7628-161X Chen Cun 3 chencun@cdnu.edu.cn
Li Zheng-hao 1
Zhao Yu-yang 3
Yang Yi 1
Liu Zhi-bin 1 liuzhibin@scu.edu.cn
1 Key Laboratory of Bio-Resources and Eco-Environment of Ministry of Education, College of Life Sciences, Sichuan University , Chengdu , China
2 Kiwifruit Breeding and Utilization Key Laboratory, Sichuan Provincial Academy of Natural Resource Sciences , Chengdu , China
3 Sichuan Provincial Key Laboratory for Development and Utilization of Characteristic Horticultural Biological Resources, College of Chemistry and Life Sciences, Chengdu Normal University , Chengdu , China
Sotelo-Mundo Rogerio
Electronic publication date: 2020 Aug 14
Publication date: 2020
Volume: 8
Electronic Location ID: e9724
Received 2020 Jan 27; Accepted 2020 Jul 23
Copyright: © 2020 Qiu et al.
Copyright year: 2020
Copyright holder: Qiu et al.
License: This is an open access article distributed under the terms of the Creative Commons Attribution License, which permits unrestricted use, distribution, reproduction and adaptation in any medium and for any purpose provided that it is properly attributed. For attribution, the original author(s), title, publication source (PeerJ) and either DOI or URL of the article must be cited.
License URL: https://creativecommons.org/licenses/by/4.0/

Keywords: ‘Hongyang’ kiwifruit, CPPU, Nutritional metabolism, Development process, Correlation analysis

Funding: Sichuan Science and Technology Program 2018NZ0044 and 2017JY0178 Sichuan College Students’ Innovation and Entrepreneurship Training Project 201814389144 This work was supported by funding from Sichuan Science and Technology Program (Nos. 2018NZ0044 and 2017JY0178) and Sichuan College Students’ Innovation and Entrepreneurship training Project (No. 201814389144). The funders had no role in study design, data collection and analysis, decision to publish, or preparation of the manuscript.

==============================
Forchlorfenuron, N-(2-chloro-4-pyridyl)-N-phenylurea (CPPU), is often used to promote fruit growth and improve production. The role of CPPU in kiwifruit growth has been established. However, the correlation between fruit weight and nutritional metabolism during development after CPPU treatments remains largely undetermined. Here, we surveyed the variations in weight and nutrient components of the ‘Hongyang’ kiwifruit (Actinidia chinensis) when CPPU was sprayed on fruit 25 days after anthesis. The CPPU application did not significantly influence the dry matter, soluble solids, starch, vitamin C or protein concentrations. However, the fresh weight, length and maximum diameter were significantly increased compared with the control. Moreover, in fruit of the same developmental stage, the fructose, glucose and soluble sugar levels increased after the CPPU treatment, compared with the control. On the contrary, citric, quinic and titratable acid concentrations decreased. However, a correlation analysis between fresh weight and the nutritional contents revealed that CPPU did not affect the concentrations of the most abundant organic acids (quinic and citric) and sugars (glucose, fructose and sucrose), compared with control fruit of the same weight. Therefore, CPPU applications enhance ‘Hongyang’ kiwifruit weight/size. However, there were no significant differences in the nutritional qualities of treated and untreated fruit having the same weights.

Introduction

Owing to its pleasant fragrance, and high carbohydrate, vitamin and folic acid concentrations, the kiwifruit has become an economically and nutritionally important fruit crop (Richardson, Ansell & Drummond, 2018). China is the largest producer of kiwifruit worldwide with a production ~2.04 million tons in 2018 (FAO stat, 2018). The green-fleshed ‘Hayward’ (Actinidia deliciosa (A. Chev.) C.F. Liang and A.R. Ferguson.) bred in New Zealand, was the first globally traded kiwifruit cultivar. Subsequently, the yellow-fleshed ‘Hort16A’ (A. chinensis Planch. var. chinensis) cultivar was selected and stimulated the development of the world kiwifruit industry. The success of ‘Hort16A’ kiwifruit is ascribed largely to the tropical-like flavors and sweeter yellow-flesh compared with ‘Hayward’ kiwifruit (Pranamornkith, East & Heyes, 2012; Du et al., 2019). In recent years, consumers have preferentially selected kiwifruit based on the flesh color. ‘Hongyang’ (A. chinensis Planch. var. chinensis), cultivated by the Sichuan Provincial Nature Resources Institute, is the first red-fleshed kiwifruit cultivar to be grown on a commercial scale (Zhang et al., 2018). ‘Hongyang’ is quite popular among consumers owing to its red inner pericarp and superior fruit properties, including better taste and higher vitamin C concentrations (He et al., 2018). Therefore, it quickly became a significant component of the Chinese kiwifruit industry.

To achieve the most desirable characteristics at harvest time, kiwifruit must attain a large well-shaped form and contain a balance of soluble sugars and organic acids (Cruz-Castillo et al., 2014). These fruit-related parameters are considered to be important in terms of quality and consumer appeal (Ainalidou et al., 2015). Therefore, fruit weight/size, and the soluble sugar and organic acid concentrations are the main parameters used to grade kiwifruit. Bioregulators, such as forchlorfenuron, N-(2-chloro-4-pyridyl)-N-phenylurea (CPPU), gibberellin, 2,4-dichlorophenoxyacetic acid (2,4-D) and thidiazuron, are often applied to improve productivity of fruit crops (Famiani et al., 2007). Recently, the synthetic cytokinin CPPU has become the most widely used owing to a strong ability to stimulate fruit growth (Cruz-Castillo et al., 2014).

At present, it is legal to use CPPU commercially for fruit production in Italy, Japan, Chile and the USA (Cruz-Castillo et al., 2014). CPPU can significantly promote cell division and enhance the sizes of some fruit, such as watermelons, apples, grapes and macadamia. However, CPPU’s effects on the fruit quality attributes were inconsistent. For example, compared with the untreated control, there were similar soluble solids concentration (SSC) and decreased total organic acid concentrations in CPPU-treated ‘Cuiguan’ pear (Niu et al., 2015). By contrast, CPPU applications reduced the SSC and enhanced the accumulation of titratable acidity (TA) in ‘Flame Seedless’ grapes (Peppi & Fidelibus, 2008). Nevertheless, Stern et al. (2003) found that CPPU applications did not produce any negative effects on the fruit quality attributes of ‘Royal Gala’ apple at harvest.

For kiwifruit, the SSC and TA level in CPPU-treated fruit of A. arguta (Sieb. et Zucc.) Planch. ex Miq. ‘Mitsuko’ were lower than those of untreated fruit (Kim et al., 2006). On the contrary, CPPU applications led to an increased SSC level and a similar TA concentration, when compared with the untreated ‘Hayward’ kiwifruit (Ainalidou et al., 2015). The reason for the different effects of CPPU on the fruit quality attributes of ‘Mitsuko’ and ‘Hayward’ remains undetermined. In the previous studies, the effects of CPPU on fruit nutrition were mainly focused on the same development stage. There was a lot of difference of fruit quality between CPPU-treated and control fruit at this point. However, the reason was mainly because the weight of CPPU-treated fruit was much more than that of control group at the same time. In our present research, we want to know whether the nutritional metabolism of fruit was affected by CPPU application at the same weight. In this study, we examined the weight and composition of ‘Hongyang’ at various developmental points after CPPU was applied. Then, the correlations between fruit weight and the major organic acid and sugar metabolic profiles were analyzed.

Materials and Methods

Field experiment

Field experiments were carried out during 2018 in Shifang County, Sichuan (31°13′N, 101°01.161′E, 662 m a.s.l.). The grafted kiwifruit vines were trained by a Pergola system with 3-m spacings within rows and 4-m spacings between rows. In total, 40 representative vines were randomly selected and labeled for the subsequent experiments. At 25 days after anthesis (DAA), half of the vines were treated with 5 ppm CPPU (Cruz-Castillo et al., 2014) for 10 s. Similar fruit growing on another 20 vines were treated with water only as the control group. During the experiment, the orchard received normal agricultural practices, and no extreme weather conditions were recorded. Kiwifruit samples were collected at 25, 40, 55, 70, 85, 100, 115 and 130 DAA. At each sampling date, 10 randomly selected fruits were harvested from six vines for per treatment and there were three replications.

Fruit samples were collected from three biological replicates, snap-frozen in liquid nitrogen, and then stored at −80 °C for subsequent analyses.

Evaluation of fruit weight, dry matter, shape and SSC

Fruit fresh weight and dry matter were measured as reported by Nardozza et al. (2010) Shape traits (length and minimum and maximum diameters) were measured using Vernier calipers. SSCs were assayed in juice extracted from both the ends of the fruit using a WYT-4 refractometer (Jingmi, Shanghai, China).

Total soluble sugars, TA, soluble protein, starch and vitamin C concentrations

Total soluble sugars was extracted using the method of Irigoyen, Einerich & Sánchez-Díaz (1992). In total, 0.5 g of fresh pulp was ground to a powder in the presence of liquid nitrogen, and then 15 mL 80% ethanol was added. Samples were centrifuged for 10 min at 7,000 rpm. Afterwards, two mL of supernatant were saved for the soluble sugar determination. Total soluble sugars were assessed using the colorimetric anthrone method (Huang et al., 2017). TA and soluble protein concentrations were determined as described in Kwon et al. (2019) and Wang et al. (2016), respectively. The precipitation obtained when extracting sugars was autoclaved for 1 h and then incubated with amyloglucosidase in 12 mL acetate buffer (0.25M, pH4.5, 55 °C) for a 1 h. The sample was then centrifuged for 10 min at 7,000 rpm. The supernatant was used to measure the starch concentration using a colorimetric determination (Smith, Clark & Boldingh, 1992). Vitamin C was measured using a 2,6-dichloroindophenol titration method (Ma et al., 2017; Alhassan et al., 2019).

HPLC analyses of organic acids and carbohydrates

Organic acids were detected according to Zheng et al. (2009). In total, 10 g kiwifruit was extracted in a 50-mL centrifuge tube containing 12.5 mL NH4H2PO4 (40 mmol/L, pH 2.5) and centrifuged at 12,000×g at 4 °C for 15 min. The supernatant was collected and subsequently filtered. An HPLC analysis was performed using an Agilent 1,200 Series instrument with a C18 column (4.6 mm × 250 mm, 5 µm; Waters, UK) at 210 nm. The mobile phase was NH4H2PO4 (40 mmol/L, pH 2.5). The sugar concentration was calculated as described in Barboni, Cannac & Chiaramonti (2010) using an Agilent 1,200 Series instrument equipped with a refractive index detector (Agilent, CA, USA), and the extract was separated using an amino column (Inertsil NH2 250 mm × 4.6 mm, 5 µm) and eluted with water (Milli-Q) at a flow rate of 0.6 mL/min. Elution was carried out isocratically for 20 min at 85 °C.

Data analyses

The experimental data are presented as the means ± standard errors from three independent experimental replications. In the analysis process, the main factor of the ANOVA is whether CPPU was added. A single factor ANOVA (ANOVA-LSD) was used to make comparisons among groups. A difference was considered statistically significant at p < 0.05. In the early stage of data collection, each sample was repeated three times. In the analysis process, taking time as a factor, whether the fruit was treated was correlated with the change in nutritional index content within equal time periods was considered. The software R 3.0.0 (R Development Core Team, 2008) and SPSS 16.0. were mainly used for the analyses. The statistical analysis was mainly performed using time as the dependent variable. For repeated tests, the average value of three repeated test results at each time point was used in the regression analysis to reduce the error caused by the use of a single test.

Results

Fruit length, diameter and weight

The application of 5 ppm CPPU increased A. chinensis ‘Hongyang’ fruit weight and size at harvest (Figs. 1A and 1B). There was no obvious difference in the fresh weights of CPPU-treated and control fruit from 25 to 55 DAA. However, the fresh weight was significantly increased by the CPPU treatment from 70 to 130 DAA. At harvest, CPPU-treated fruit had a final fresh weight that was, on average, 14% greater than that of the control (Fig. 1B).

Figure 1 Effects of CPPU on the size (A), fresh weight (B), length (C), maximum diameter (D), minimum diameter (E) of ‘Honyang’ Kiwifruit.

CK, control; CPPU, CPPU-treated fruit; DAA, days after anthesis. Each value represents the mean of three biological replicates of 10 fruit each, *p < 0.05, **p < 0.01.

Compared with untreated fruit, the CPPU-treated fruit had greater fruit lengths and maximum diameters by 8.2% and 7%, respectively, at harvest (Figs. 1C and 1D). The length was significantly increased from 70 to 130 DAA, and the maximum diameter was enhanced from 55 DAA until the fruit was harvested. However, the minimum fruit diameter was not influenced by CPPU application during kiwifruit development (Fig. 1E).

Fruit dry matter and soluble solids concentration

The dry matter and SSC were not affected by CPPU treatments. The percentage of fruit dry matter increased during fruit development as expected (Fig. 2A). The SSC did not change from 25 to 100 DAA. However, there was a dramatic growth from 115 DAA to 130 DAA, with the maximum SSC occurring at harvest in both treated and untreated fruit (Fig. 2B).

Figure 2 Effects of CPPU on the dry matter (A) and soluble solid (B) of ‘Hongyang’ kiwifruit.

CK, control; CPPU, CPPU-treated fruit; DAA, days after anthesis. Each value represents the mean of three biological replicates of 10 fruit each.

Soluble sugar, TA, vitamin C, starch and protein

The application of CPPU elevated fruit soluble sugar concentrations. Soluble sugar accumulation was low in both treated and untreated fruit from 25 to 100 DAA. After 100 DAA, there was a significant increase in the soluble sugar level. Moreover, the soluble sugar concentration in CPPU-treated fruit was greater than in untreated fruit (Fig. 3A). At harvest, CPPU applications resulted in a ~14.5% increase in soluble sugars compared with the untreated fruit.

Figure 3 Effects of CPPU on the soluble sugar (A), titratable acid (B), ascorbic acid (C), starch concentration (D) and protein concentration (E) of ‘Hongyang’ kiwifruit.

CK, control; CPPU, CPPU-treated fruit; DAA, days after anthesis. Each value represents the mean of three biological replicates of 10 fruit each, *p < 0.05.

On the other hand, CPPU did not significantly affect the TA level from 25 to 100 DAA; however, the effect was obvious from 115 DAA until fruit harvest (Fig. 3B). Where, the application of CPPU led to a decline of ~6.4% in the TA concentration, compared with the untreated fruit. Meanwhile, the vitamin C, starch and soluble protein were not significantly influenced by CPPU. The vitamin C concentration decreased (Fig. 3C), while the starch and protein concentrations increased during ‘Hongyang’ kiwifruit development (Figs. 3D and 3E).

Sugars and organic acids

N-(2-chloro-4-pyridyl)-N-phenylurea applications significantly increased fructose and glucose concentrations. Differences in fructose levels between treated and untreated fruit were first detectable at 70 DAA and continued until harvest. In addition, from 85 to 130 DAA, the glucose concentration of the CPPU-treated fruit was much greater than that of the untreated fruit. By harvest time, the CPPU applications had significantly increased the fructose and glucose concentrations in fruit by 75.2% and 43.1%, respectively, compared with untreated fruit (Figs. 4A and 4B). On the contrary, the CPPU application decreased the sucrose concentration. The effect was evident in fruit from 85 DAA until final harvest. Finally, the sucrose concentration of CPPU-treated fruit was significantly lower by approximately 23.5%, compared with the concentration of untreated fruit (Fig. 4C).

Figure 4 Effects of CPPU on the fructose (A), glucose (B), sucrose (C), quinic acid (D), citric acid (E) and malic acid (F) concentration of ‘Hongyang’ kiwifruit.

CK, control; CPPU, CPPU-treated fruit; DAA, days after anthesis. Each value represents the mean of three biological replicates of 10 fruit each, *p < 0.05, **p < 0.01.

The quinic acid concentration of fruit declined after CPPU applications. There was a difference in the quinic acid concentration at 70 DAA until final harvest. Finally, the CPPU application caused a decrease of ~19.5% in quinic acid compared with the untreated fruit (Fig. 4D). The citric acid concentration was not significantly affected by CPPU from 25 to 115 DAA. However, at 130 DAA, the citric acid level in the control fruit was significantly greater than that of CPPU-treated fruit. When the fruit was harvested, the citric acid concentration in untreated fruit was ~18% greater than that in CPPU-treated fruit (Fig. 4E). For malic acid, CPPU applications did not significantly influence the level, compared with the control fruit (Fig. 4F).

The correlations between fruit weight and nutritional parameters

For the fruit sample data, the log function was used to linearize the glucose concentration distribution and a linear regression equations of this fruit nutrient as well as the others and weight were obtained. The regression results are shown in Figs. 5A–5E and Table 1, and almost all the results had good degrees of fit (R2 > 0.8). For sugars, there were no significant differences in the slope and intercept between CPPU-treated and control fruit. Using the same statistical analysis method, quinic acid and citric acid had values greater than 0.05 for both the slopes and intercepts.

Figure 5 Regression analysis between weight and the quinic acid (A), citric (B), fructose (C), glucose (D) and sucrose (E) concentration of ‘Hongyang’ kiwifruit.

CK, control; CPPU, CPPU-treated fruit.

Table 1 Weight and the nutritional parameters regression analysis for CPPU-treated and control group fruit.

	Group	Model	Model Comparisons (P-values)	
		Equation	R2	Slope	Intercept	
Glucose	CK	Y = 0.0423x + 0.2029	0.965	0.526	0.828	
	CPPU	Y = 0.0386x + 0.2787	0.954			
Fructose	CK	Y = 0.0343x + 0.7134	0.974	0.915	0.694	
	CPPU	Y = 0.0348x + 0.6323	0.978			
Sucrose	CK	Y = 0.0308x + 0.6421	0.961	0.021	0.887	
	CPPU	Y = 0.0241x + 0.6982	0.907			
Quinic acid	CK	Y = −0.0091x + 1.4482	0.883	0.114	0.163	
	CPPU	Y = −0.0119x + 1.6152	0.612			
Citric acid	CK	Y = 0.0108x − 0.1553	0.876	0.103	0.956	
	CPPU	Y = 0.008x − 0.1174	0.918			
Note:

CK, Control; CPPU, CPPU-treated fruit.

Discussion

Fruit weight/size and dry matter are important integrated indices used to evaluate fruit quality (Jaeger et al., 2011). In this study, the fruit weight curve of ‘Hongyang’ kiwifruit was similar with that of ‘Hayward’ kiwifruit. There was a rapid development from 25 to 55 DAA (Fig. 1B), and fresh weight continued to rapidly increased in the ‘Hayward’ kiwifruit from 35 to 75 DAA (Nardozza et al., 2017). After CPPU application, the fresh fruit weight significantly increased (Fig. 1B). Similar results were also reported in the ‘Hayward’ kiwifruit after the 10 mg/L CPPU treatment at 15 days after fruit set, whereas the ‘Hongyang’ kiwifruit weight increase was small compared with that of the ‘Hayward’ kiwifruit, which increased by ~30% (Ainalidou et al., 2015). In contrast, the increases in the dry matter were only slightly different between the ‘Hongyang’ and ‘Hayward’ kiwifruit when 10 ppm CPPU was applied at 28 DAA (Nardozza et al., 2017). The dry matter dramatically increased from 40 to 55 DAA in ‘Hongyang’ kiwifruit, while the rate of increase in ‘Hayward’ was basically constant. At harvest, the dry matter was not affected by CPPU treatments (Fig. 2A), which agreed with the data from CPPU-treated ‘Hayward’ kiwifruit when 4 ppm CPPU was sprayed ~1 week before anthesis (Cruz-Castillo et al., 2014).

The fruit length and diameter define the shape of the fruit. CPPU applications promote cell division and/or expansion to increase fruit size (Lewis et al., 1996). The length curve of ‘Hongyang’ kiwifruit presented as single sigmoidal in shape (Fig. 1C), and it was different from that of ‘Mitsuko’ kiwifruit receiving the 5 mg/L CPPU treatment at 10 days after petal fall, which showed a double sigmoidal curve (Kim et al., 2006). The shape of CPPU-treated ‘Hongyang’ kiwifruit was a little thinner at the blossom end, compared with untreated fruit (Figs. 1C and 1D).

Soluble solids concentration is a crucial factor for evaluating the fruit ripening time (Koutsoflini, Gerasopoulos & Vasilakakis, 2013). No significant differences in SSC values were observed between untreated and CPPU-treated fruit (Fig. 2B). SSC refers to all compounds dissolved in water, including sugar, acid, fibrin and mineral components, although the CPPU application increased the fruit soluble sugar concentration, it did not affect the SSC. This may result from the decreases in sucrose, quinic acid, citric acid and other soluble substances. Similar results and growth patterns were observed by Iwahori, Tominaga & Yamasaki (1988) in 40 mg/L CPPU-treated ‘Hayward’ kiwifruit.

Sugars and organic acids are the parameters of greatest concern regarding the flavor of fruit (Nishiyama et al., 2008; Richardson et al., 2019). Compared with kiwifruit controls, the soluble sugar concentrations increased 23% by a 20-ppm CPPU application at 14 DAA (Famiani et al., 1999), and TA level decreased 13.2% when 10 ppm CPPU was applied at 30 days after petal fall (Pramanick et al., 2015). However, the concentrations of soluble sugars in CPPU-treated ‘Hongyang’ kiwifruit increased by ~8.8% and TA decreased by 4.4% compared with in untreated fruit. Thus, the effects of CPPU on the taste-related characteristics of the ‘Hongyang’ kiwifruit were more limited than on other kiwifruit cultivars, this may be the result of different concentrations and times of CPPU applications.

The vitamin C concentration is a preferential factor for the assessment of the quality levels of many fruit. Kiwifruit contains more ascorbic acid than oranges, strawberries, lemons and grapefruit (Ma et al., 2017). Here, there were no differences in the vitamin C concentration between CPPU-treated and untreated fruit, which was consistent with the findings in cucumber (Qian et al., 2018). However, 5 mg/L CPPU treatment at 25 days after petal fall, the vitamin C concentration in A. argute kiwifruit was less, than that of untreated fruit (Kim et al., 2006).

The changes that occur in protein levels during kiwifruit development are unknown. In the present study, the protein concentration showed similar values between CPPU-treated and untreated fruit (Fig. 3E). Qian et al. (2018) also reported that the protein concentration of cucumber was not affected by CPPU.

The accumulation of starch was similar to that of the ‘Hayward’ kiwifruit (Nardozza et al., 2017). However, the ‘Hayward’ kiwifruit had its maximum starch concentration at 125 DAA, while in ‘Hongyang’ kiwifruit this occurred at 100 DAA. There have been contradictory results on the effects of CPPU on the starch concentrations. The CPPU treatment was reported to significantly reduce the starch concentration of ‘Hayward’ kiwifruit (Nardozza et al., 2017). However, it has also been reported that 20-ppm CPPU-treated ‘Hayward’ kiwifruit at 14 DAA had higher starch concentrations than untreated fruit (Antognozzi et al., 1996). The current data indicated that CPPU applications did not result in a significant change in the starch concentration of ‘Hongyang’ kiwifruit (Fig. 3D). The results were consistent with other findings in ‘Hayward’ kiwifruit (Cruz-Castillo et al., 2014). After the kiwifruit matures, the starch degrades into soluble sugar. However, the concentrations of starch and soluble sugar were increased in the developmental processes. In addition, the changes in starch and soluble sugar were not consistent (Nardozza et al., 2010). Moreover the soluble sugars content of CPPU-treated fruit were higher than that of control in Actinidia deliciosa, while the starch content was similar to the control at harvest (Antognozzi et al., 1996).

The most abundant sugars in ‘Hongyang’ kiwifruit are glucose, fructose and sucrose. Citric, quinic and malic acids are the main organic acids (Nishiyama et al., 2008). Compared with the control, at the same developmental stages, there were remarkable increases in fructose and glucose levels (Figs. 4A and 4B), but the sucrose concentration declined at harvest after fruit was CPPU treated (Fig. 4C). Similar results were reported in the CPPU-treated ‘Hayward’ kiwifruit (Nardozza et al., 2017). However, there was a difference in the accumulation of glucose between ‘Hongyang’ and ‘Hayward’ kiwifruit. The glucose concentration reached its maximum value at ~50 DAA in ‘Hayward’ kiwifruit, while the glucose level continued to increase in ‘Hongyang’ kiwifruit. A similar result was reported by Richardson et al. (2011) in ‘Hort16A’.

According to the regression analysis between weight and major organic acid/sugar concentrations (Fig. 5), there were no significant differences in the slopes and intercepts (Table 1) for fructose, glucose, citric acid and quinic acid between the control and CPPU-treated groups. Those findings suggested that when the ‘Hongyang’ kiwifruit developed to the same weight, the nutrition indices (fructose, glucose, citric acid and quinic acid) were similar between untreated and CPPU-treated fruit. Therefore, although there were some effects on TA and soluble sugars in CPPU-treated ‘Hongyang’ kiwifruit by comparison with untreated fruit at the same time, when the fruit was grown to the same weight, CPPU treatments did not influence the nutritional components of ‘Hongyang’ kiwifruit.

Conclusions

A CPPU application significantly altered ‘Hongyang’ kiwifruit development, leading to increases in fresh weight and sugar concentrations, as well as a decline in organic acid concentrations at the same developmental stage as the untreated kiwifruit. However, the impacts on ‘Hongyang’ were lesser compared with other kiwifruit cultivars treated with CPPU. In contrast, from the regression analysis between weight and nutritional indices, the CPPU treatment did not affect major organic acid and sugar concentrations considering a constant fruit weight. These results indicated that CPPU treatments have the potential to increase ‘Hongyang’ kiwifruit production with limited negative effects on quality.

Supplemental Information

Supplemental Information 1 Raw data of fresh weight.

The HY represents untreated fruit, the HYP represents CPPU-treated fruit.

Click here for additional data file.

Supplemental Information 2 The raw data of maximum diameter, minimum diameter and length.

The HY represents untreated fruit, the HYP represents CPPU-treated fruit.

Click here for additional data file.

Supplemental Information 3 The raw data of dry matter.

The raw data of dry matter. The HY represent untreated fruit, the HYP represent CPPU-treated fruit.

Click here for additional data file.

Supplemental Information 4 The raw data of soluble solids.

The HY represents untreated fruit, the HYP represents CPPU-treated fruit.

Click here for additional data file.

Supplemental Information 5 The raw data of soluble sugar.

The HY represents untreated fruit, the HYP represents CPPU-treated fruit.

Click here for additional data file.

Supplemental Information 6 The raw data of titratable acid.

The raw data of titratable acid. The HY represent untreated fruit, the HYP represent CPPU-treated fruit.

Click here for additional data file.

Supplemental Information 7 The raw data of vitamin C.

The HY represents untreated fruit, the HYP represents CPPU-treated fruit.

Click here for additional data file.

Supplemental Information 8 The raw data of starch content.

The HY represents untreated fruit, the HYP represents CPPU-treated fruit.

Click here for additional data file.

Supplemental Information 9 The raw data of protein content.

The HY represents untreated fruit, the HYP represents CPPU-treated fruit.

Click here for additional data file.

Supplemental Information 10 The raw data of fructose, glucose, sucrose.

The HY represents untreated fruit, the HYP represents CPPU-treated fruit.

Click here for additional data file.

Supplemental Information 11 The raw data of quinic acid, citric acid and malic acid.

The HY represents untreated fruit, the HYP represents CPPU-treated fruit.

Click here for additional data file.

Additional Information and Declarations

Competing Interests

Author Contributions

Data Availability

The authors declare that they have no competing interests.

Guo-liang Qiu conceived and designed the experiments, performed the experiments, prepared figures and/or tables, and approved the final draft.

Qi-guo Zhuang conceived and designed the experiments, performed the experiments, prepared figures and/or tables, and approved the final draft.

Yu-fei Li performed the experiments, authored or reviewed drafts of the paper, and approved the final draft.

Si-yu Li analyzed the data, authored or reviewed drafts of the paper, and approved the final draft.

Cun Chen conceived and designed the experiments, analyzed the data, prepared figures and/or tables, authored or reviewed drafts of the paper, and approved the final draft.

Zheng-hao Li performed the experiments, analyzed the data, authored or reviewed drafts of the paper, and approved the final draft.

Yu-yang Zhao performed the experiments, analyzed the data, authored or reviewed drafts of the paper, and approved the final draft.

Yi Yang conceived and designed the experiments, analyzed the data, authored or reviewed drafts of the paper, and approved the final draft.

Zhi-bin Liu conceived and designed the experiments, analyzed the data, prepared figures and/or tables, and approved the final draft.

The following information was supplied regarding data availability:

The raw data are available in the Supplemental Files.

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
