# Peer review of "Correlation between fruit weight and nutritional metabolism during development in CPPU-treated Actinidia chinensis ‘Hongyang’"

_PeerJ, doi:10.7717/peerj.9724_

## Round 0.1 · original submission · Major Revisions

As the reviewer´s suggested, professional proofreading service is advisable to improve the quality of the manuscript.

Reviewer 1 ·

Basic reporting

Please add more relevant recent papers.

Experimental design

Please add more relevant recent papers.

Validity of the findings

Please add more relevant recent papers.

Additional comments

Please add more relevant recent papers.

Reviewer 2 ·

Basic reporting

Although this research aims an original topic trying to relate the content of sugars and organic acids with the weight of kiwi fruit by the effect of the preharvest application of a growth regulator, the main research question is not well supported.

Experimental design

Even when the introduction makes clear that the CPPU regulator influences the development and metabolism of kiwifruit, it is not clear why it is important to study the relationship between fruit weight and the content of a particular component.

The implemented techniques are not the more advanced but they are adequate for the proposed response variables.

Validity of the findings

The manuscript is well written, the results seem sound, discussion and interpretation are sober and supported by their results, however to my respect the generated knowledge may not be relevant enough to the field and there is not a clear meaningful contribution to be considered for publication in PeerJ.

Reviewer 3 ·

Basic reporting

English needs substantial editing.
some references missing, some not appropriate (See details below).
Figures need improvement (some suggestions below)
Raw data shared.

Experimental design

The statistical analysis is not very clear and the experimental design dubious. Not sure if what it is called a biological replicate is a true biological replicate or more of a technical replicate given that the experimental unit was not really defined. Some graphs, where stars show statistically significant differences the error bars between the treatment and the control overlap, raising concerns about how the statistical analysis was performed.
Methods need more information (see detailed suggestions).
Research questions at the end of the introduction not very clear (see detailed suggestions)

Validity of the findings

The authors have assessed the effect of low dosage CPPU (5 ppm) on ‘Hongyang’ kiwifruit applied at 25 days after anthesis. Fruit were 14% larger. SSC was not affected by CPPU but TA decreased in treated fruit. Analysis of soluble sugars revealed that CPPU increased hexoses (fructose and glucose) but lowered sucrose. This is in disagreement with the SSC data, but the authors do not discuss it in the manuscript. TA and acids are in agreement.
Discussion does not consider factors such as CPPU concentration applied and timing of application. this should be added when drawing conclusions as it has been clearly shown how different timing of application in 'Hayward' have different outcomes.

Additional comments

Some specific comments:
Line 22: …to influence post-harvest quality. Not sure this is the reason why it is used. it is more a consequence of its use.
Lines 44-45: Guo et al 2017 reports the figure of 1.06 million tons per year, but they do not have a citation for it. What is the source of this production figure?
Line 46: kiwifruit species have been recently reclassified. Actindia chinensis var. deliciosa (A.Chev.) A.Chev. is the species for ‘Hayward’ and A. chinensis Planch. var. chinensis is the species for ‘Hort16A’ and ‘Hongyang’. Please check manuscript and edit accordingly. Actinidia can be abbreviated after first mention. Authority to be added to materials and methods section only. Check spelling for species name Actinidia arguta throughout the manuscript.
Line 54: red fleshed core? ‘Hongyang’ core is white. It is the locules tissue where the seeds are (inner pericarp) that accumulates anthocyanin. The outer pericarp is green or yellow according the level of degreening that occurred and the maturity/ripening stage. Please amend accordingly.
Line 54: check text to make sure ‘Hongyang’ (and any other cultivar name) is between single quotes and not between double quotes or single quotes are missing.
Line 55: the kiwifruit in Henwood et al 2018 is not ‘Hongyang’ (star-red fleshed) but is a block-red fleshed fruited genotype. therefore this reference is not appropriate here.
Lines 62-65: references missing, please add.
Line 67: nutritional qualities? Please change to fruit quality attributes, here and for the rest of the manuscript as this terminology is more appropriate.
Lines 80-82: what’s your hypothesis? It is not really clear from this paragraph. add some more rather than only have we have tried this product on a different cultivar and checked the effects.
Line 87: were the kiwifruit vines grafted or were they on their own roots? Please specify. Also this sentence needs English editing.
Line 90: unsure why you cite this paper here (Cruz-Castillo 2014) as they did pre-flowering CPPU treatments. Were just the CPPU exposure and the concentration used that were similar?
Line 90: half of the fruit were dipped. This is unclear. I think the author, reading the text that follows, meant that they treated half of the vines with CPPU and half of the vines were control.
Line 91: kiwifruit plants are called vines and not trees, as they are botanically vines and not trees. Please check the manuscript for consistency in terminology making sure you do not use the word tree.
Line 94: the experimental design is vague and I am not satisfied the ‘biological replicates’ are really independent. The author state that 30 fruit were randomly collected and then subdivided into 3 biological replicates. To me you have one biological replicate and 3 technical replicates. The vines should have been subdivided into 3 groups and fruit from each group treated as a biological replicate.
Line 102: please make sure that all equipment cited in the paper has maker’s name, city and country.
Line 104: there are no details on how sugars and starch were extracted (only details on how they were determined). What’s the method used? please describe. It is also important to describe how samples were handled (how were they ground, were they kept frozen during grinding?)
Lines 112 and 114: is the pH of the NH4H2PO4 solution 2.5 or 2.54? please check you have the correct figure there.
Line 117: please delete Toussant as it is Barboni’s first name.
Line 123: I do not think the three replicates were true biological replicates and independent from the description given in the materials and methods given that they were randomly harvested and the grouped later.
Line 124: you need more details about the ANOVA. Did you consider you had repeated measures into your statistical analysis? You are analysing a time course. Add more details. What software did you use for the anova? Which are the factors? Is time a factor in your analysis? How did you treat the replicates?
Line 220: kiwifruit protein influencing fruit quality? The only protein that really affects kiwifruit quality is actinidin as it can trigger allergies in some individuals. Either add references in this part or remove the protein discussion point as it is not relevant for this paper.
Line 224: the statement about starch digestibility is not appropriate here. in kiwifruit starch is accumulated during development as a strategy to store carbon without affecting the osmotic pressure of the fruit and maximise its sink capacity. When eating ripe, starch is not there as it has been converted in soluble sugars. Please remove sentence.
Figure 3 shows that CPPU treated fruit have higher soluble sugars (%). This is because CPPU advances fruit maturity. How do you relate this to unchanged SSC in Figure 2? How would the author explain no differences in SSC, but higher soluble sugars and no change in starch? Starch is the ‘carbohydrate’ long term storage in kiwifruit fruit. A comment about this point in difference in ‘Hongyang’ might be needed in the discussion. Also, timing of application (After anthesis) and concentration of CPPU applied should be factored in. Also, what was the crop load like? If the crop load was quite low, then it is likely that vines were not carbohydrate limited and tolerated the moderate increase in fruit size (only 14%) better than other conditions when the increase in size was far more dramatic.
Figure 1: what is DAA? I assume days after anthesis. what is CK? I assume it is untreated control, but it is not stated in the figure legend of any of the figures (this comments apply to all of the figures). The figure legends should allow figure interpretation without the rest of the manuscript (Stand alone). Please revise all of the figure legends to make sure that sufficient details are provided. X-axes: please add a title for it. example: Developmental stage (DAA). This comment also applies to all of the graphs in the manuscript.
Figure 2 is deformed and needs fixing. Also, comments that applies to other graphs. Please have the X-axes and the Y-axes starting from zero. Interruptions of axes are useful when you have a large increase in a particular parameter you are measuring, not for a few units. Please remove when not required to improve clarity.
All figures and text: when talking about metabolites, you have to refer to it as concentration not content. E.g. Sucrose concentration (mg/100g).
Are the concentration in mg/100g of fresh weight? If so specifiy.
References: please check them accurately for spelling, capitalisation, italic etc…

---

## Round 0.2 · Minor Revisions

Please take into consideration the reviewer's comments and the attached PDF with detailed comments and corrections. After addressing all the comments, please provide a revised manuscript and a detailed point-by-point rebuttal letter.

Reviewer 3 ·

Basic reporting

I am satisfied with the changes

Experimental design

I am satisfied with the changes

Validity of the findings

I am satisfied with the changes

Reviewer 4 ·

Basic reporting

A professional English is used in most of the manuscript, some details that need edition are suggested and marked at the reviewed pdf file. The title could be edited to say: "Correlation between fruit weight and nutritional metabolism during development in CPPU-treated Actinidia chinensis ‘Hongyang’".
References were added by the authors according to the suggestion previously made. However, two references are misplaced at the materials and methods section, they are marked at the pdf file of the manuscript reviewed.
Figures are professional and a few suggestions are given at the pdf reviewed to make figure legend shorter at Y axis.
Results are well presented and some suggestions are made at the pdf manuscript to make them clearer and a few questions are made to the authors.

Experimental design

The manuscript presents an original research to determine the effects on physiological and nutritional contents of Actinidia chinensis fruit during development, due to the application of a fruit growth promoter. The addressed question is stated in the manuscript. Methods are referenced to published papers. Three biological replicates of 10 fruit each were used to perform the experiments. Comparisons with other Actinidia chinensis that used the same growth promoter are made in the research presented. A correlation analysis of fruit weight with quality parameters is presented. Besides effects on sugars and organic acids after the treatment are smaller when compared with other cultivars.

Validity of the findings

The benefit of assaying different concentrations of the fruit growth promoter is stated in the paper and compared to other kiwi cultivars. The statistics are sound and discussed in the paper. Conclusions are based on the data presented and discussed.

Additional comments

The paper presents a sound work and some suggestions are made on the pdf to make it clearer. Two references were switched with respect to the methods used to determinate TA and soluble protein. The Y axis at figures could be shortened as suggested. Some suggestions are made throughout the manuscript and also for the title. I hope those suggestions are helpful to you.

Annotated reviews are not available for download in order to protect the identity of reviewers who chose to remain anonymous.

---

## Round 0.3 · Minor Revisions

The manuscript has improved over the review rounds and it is now almost ready to be accepted at PeerJ.

Before that, please address the following from the Section Editor, Gerard Lazo:

He asks that you either provide a pointer to the Pesticide Fact Sheet because this is a chemical applied to food product to assure the reader the unremarkable toxicity of the compound on food as the chemical is the key subject matter; or maybe to specifically mention that the application is safe and acceptable, or to clarify more that this is a standard practice in treatment that end in a product for human consumption somewhere.

---

## Round 0.4 · accepted · Accept

Thanks for including information regarding the biosafety on CCPU.